# Sub-Task Discovery with Limited Supervision: A Constrained Clustering Approach

**Phillip Odom, Aaron Keech & Zsolt Kira**
Georgia Tech Research Institute
Atlanta, GA 30332, USA
phodom@gatech.edu, aaron.keech@gtri.gatech.edu, zkira@gatech.edu

## Abstract

Hierarchical reinforcement learning captures sub-task information to learn modular policies that can be quickly adapted to new tasks. While hierarchies can be learned jointly with policies, this requires a lot of interaction. Traditional approaches require less data, but typically require sub-task labels to build a task hierarchy. We propose a semi-supervised constrained clustering approach to alleviate the labeling and interaction requirements. Our approach combines limited supervision with an arbitrary set of weak constraints, obtained purely from observations, that is jointly optimized to produce a clustering of the states into sub-tasks. We demonstrate improvement in two visual reinforcement learning tasks.

Sequential decision-making problems are an important domain within artificial intelligence. With complex tasks, it is necessary to model such tasks in a hierarchical manner, consisting of a set of sub-tasks that capture sub-goals. Hierarchical methods have a rich history in artificial intelligence (Sutton et al., 1999; Mehta et al., 2008; Vezhnevets et al., 2017; Banihashemi et al., 2018), and approaches for state abstractions result in more generalizable, efficient solutions.

While there are a variety of approaches using state abstraction (Bacon et al., 2017; Florensa et al., 2018; Murali et al., 2016; Hamidi et al., 2015), they require several assumptions. For example, there are several methods that use some (potentially weak) sub-task labels in order to model the decomposition. Other methods strive to find certain classes of sub-tasks or sub-task separators (e.g. bottlenecks (McGovern & Barto, 2001; Mannor et al., 2004; Simsek et al., 2005)), which limits their generality. Recent approaches such as generative models for curriculum learning (Florensa et al., 2018) or hierarchical reinforcement learning, on the other hand, require interaction or simulators that can perform roll outs in the environment.

We formulate sub-task discovery as a *constrained clustering* (Wagstaff & Cardie, 2000; Basu et al., 2008) problem, where limited supervision is combined with an arbitrary set of weak constraints obtained purely from observations and jointly optimized to produce a distinct clustering of the states into sub-tasks. These weak constraints are purely unsupervised, assuming no sub-task labels, and do not require any simulators. Specifically, we leverage recent advancements in deep learning-based constrained clustering (Hsu & Kira, 2016; Hsu et al., 2019) and show that we are able to optimize over a set of noisy weak pair-wise constraints between states (representing noisy estimates of whether two states are similar or dissimilar, i.e. belong to the same sub-task or not).

While previous work has utilized temporal information to generate constraints over objects in video (Wu et al., 2013), we explore a more general set of unsupervised constraints that can be learned in decision-making tasks. Specifically, we demonstrate two examples of constraints: 1) local constraints that capture temporal information, representing whether sequences of states belong in the same sub-task, and 2) global constraints that capture longer range similarity obtained by utilizing policy features as well as a trained autoencoder to compute distances between states.

We make the following contributions: 1) We propose a general framework that combines weak evidence via constraints in a manner that is both scalable and end-to-end, 2) We define a novel way to learn weak constraints in decision-making tasks that can be automatically generated from observations, and 3) We demonstrate that the approach can work across two complex visual environments.

CONSTRAINED SUB-TASK DISCOVERY

Sequential decision-making problems involve an agent interacting in the environment by iteratively observing the current state of the world $s_t$ and selecting actions $a_t$. A trajectory is a sequence of observations and actions over time, $s_0, a_0, s_1, a_1, ...a_{n-1}, s_n$. Trajectories may involve repeated tasks in complex environments. Representing the task structure of a domain yields significant gains when learning a policy. However, it is challenging to discover this task structure with limited labels.

The key feature of our Constrained Sub-task Discovery (CSD) approach is that it learns from limited (or no) labeled data by generating weak constraints that do not require sub-task labels or access to a simulator for evaluation. Thus, CSD, learns pairwise constraints only from the observations of the agent's trajectories. The pairwise similarity constraints intuitively represent whether two observed states belong in the same sub-task (i.e., similar) or different sub-tasks (i.e., dissimilar). CSD generates different types of weak constraints that capture the relationships between states in a trajectory and states across trajectories. It jointly optimizes over these constraints, yielding a clustering or sub-task assignment. We first describe the method for generating constraints and then discuss how the constraints are naturally combined in the constrained clustering algorithm.

**Constraint Generation**
In order to effectively generate similarity constraints over the trajectories, CSD requires features that capture the agent's behavior. Thus, we use imitation learning, which directly learns the distribution from states to actions using the agent's trajectories, to approximate the agent's policy ($\pi$). While we capture this policy with Long Short Term Memory (LSTM) (Hochreiter & Schmidhuber, 1997), which utilizes a hidden state $\mathbf{h}$ to retain memory from the trajectory, any imitation learning approach that generates intermediate features can be used to approximate this policy.

The LSTM representation, $\mathbf{h}_t$, at time $t$ depends on the current state $s_t$ and the LSTM representation at the previous state $\mathbf{h}_{t-1}$. The policy is learned over this representation.

$$\mathbf{h_t} = LSTM([\mathbf{h_{t-1}}, s_t]) \qquad P(a|\mathbf{h}_t) \approx \pi(s_t)$$

CSD uses $\mathbf{h}$ and $P$ as well as a trained autoencoder to generate different types of constraints over the trajectories ($\mathbf{c}$). We now describe the generation of our temporal and state-based constraints.

**Temporal Constraints (Local)**
Temporal constraints, $\mathbf{c}^{tc}$, intuitively capture the idea that states occurring close in time are more likely to be part of the same sub-tasks. Note that these local constraints are only created across states within a single episode. Instead of creating constraints across the entire trajectory, our approach approximates changes (or transitions) in sub-task using the learned expert policy. Inspired by traditional approaches (McGovern & Barto, 2001), we use entropy to find states with high uncertainty. We use this signal to approximate *bottleneck* states that have been traditionally identified as useful sub-goals (or transitions between sub-tasks).

First, CSD identifies bottleneck states $\mathbf{B} = \{s_{b_0}, s_{b_1}, ..., s_{b_m}\}$ where the entropy of the learned policy is greater than threshold $\tau^{tc}$ and highest within a sliding window:

$$H(P(\mathbf{h}_{s_t})) \geq \max\{\tau^{tc}, \max_{\mathbf{s_w} \in \mathbf{W}} H(P(\mathbf{h}_{s_w}))\}$$

where $\mathbf{W}$ represents the states in the sliding window and $H$ is entropy, $H(P(\mathbf{h}_s)) = -\sum_{a \in A} P(a|\mathbf{h}_s) \log P(a|\mathbf{h}_s)$.

$\mathbf{B}$ contains the states with the highest level of uncertainty. We use $\mathbf{B}$ to generate the following similarity constraints locally within a single episode[1]

$$\mathbf{c}_{ij}^{tc} = 1 \quad \forall b_d \in \mathbf{B} \text{ s.t. } d < i, j \text{ or } i, j < d, \qquad \mathbf{c}_{ij}^{tc} = -1 \qquad \exists! \, b_d \in \mathbf{B} \text{ s.t. } i < d < j \text{ or } j < d < i$$

where $\mathbf{c}_{ij}^{tc} = 1$ when states $s_i$ and $s_j$ are similar and $\mathbf{c}_{ij}^{tc} = -1$ when they are dissimilar. Intuitively, similarity constraints are generated among states that lie *between* bottleneck states, i.e., there is no bottleneck state between them. Dissimilar constraints are generated among states that are separated by a single bottleneck.

**State-based Constraints (Global)**
State-based constraints, $\mathbf{c}^{sc}$, capture both repeated sub-tasks in a single episode as well as similar

---
[1] $\exists!$ represents that there exists one and only one.

sub-tasks across episodes by identifying states that share similar features. Features can be extracted in multiple ways. We utilize two types of features: 1) $\mathbf{h}$ from the expert policy learned through imitation, and 2) $\mathbf{e}$ from an autoencoder applied to the visual states.

While it is possible to use distances in features space to generate state-based constraints, we cluster over the feature values as an intermediate step. These clusters provide a potentially more complex set of constraints than distance alone. Thus, we first cluster the states where $m_i$ represents the cluster for $s_i$. Then, CSD creates the following constraints for each set of features (creating $\mathbf{c}_{ij}^{sc_h}$ and $\mathbf{c}_{ij}^{sc_e}$).

$$\text{if } m_i = m_j, \quad \mathbf{c}_{ij}^{sc} = 1 \qquad \text{else} \qquad \text{if } m_i \neq m_j, \quad \mathbf{c}_{ij}^{sc} = -1$$

where the constraints represent whether or not states were clustered together. While these are weak constraints, we demonstrate empirically that there is value in integrating different weak constraints.

**Constrained Clustering**
We now discuss how the constraints can be utilized for sub-task discovery and how different types of constraints are effectively integrated in one framework. CSD clusters using a neural network ($f$) that takes the observation over the current state as input and outputs a distribution over the clusters.

Similar to Hsu et al. 2019, we train this clustering network using loss function $L_k$ for a set of constraints $k$.

$$L_k = -\sum_{ij} k_{ij} \cdot log(\hat{k_{ij}}) + (1 - k_{ij}) \cdot log(1 - \hat{k_{ij}}) \tag{1}$$

where $\hat{k_{ij}}$ is the probability that $s_i$ and $s_j$ are predicted to be in the same sub-task, $f(s_i) \cdot f(s_j)^T = \hat{k_{ij}}$. Thus, $L_k$ leverages the constraints by maximizing the likelihood that similar states have the same predicted cluster and minimizing the likelihood for dissimilar states.

Finally, we combine the loss w.r.t. each type of constraint. This includes the weak constraints as well as any expert-specified constraints ($\mathbf{c}^{exp}$).

$$L = \sum_{ij} w_{\mathbf{c}^{tc}} \cdot L_{\mathbf{c}_{ij}^{tc}} + w_{\mathbf{c}^{sc_h}} \cdot L_{\mathbf{c}_{ij}^{sc_h}} + w_{\mathbf{c}^{sc_e}} \cdot L_{\mathbf{c}_{ij}^{sc_e}} + w_{\mathbf{c}_{ij}^{exp}} \cdot L_{\mathbf{c}_{ij}^{exp}} \tag{2}$$

where $w_{\mathbf{c}^k}$ represents the weight applied to constraints $\mathbf{c}^k$. The weighted combination allows the prioritization of different types of constraints. We now demonstrate value of the different types of constraints empirically.

EXPERIMENTS

We evaluate our CSD approach in two visual VizDoom (Kempka et al., 2016) scenarios. Both scenarios have states that are first-person images. We now describe each domain in more detail.

The goal of the **Pickup** domain is to collect the different colored orbs that spawn in each of the four corners of the domain. The expert selects from 4 strategies for picking up orbs corresponding to 4 different pickup orders. The agent can move forward, turn left, and turn right. Collecting each orb is a different sub-task. The **Maze** domain is a navigation task in a fixed maze, where the goal is to reach the health pack at the end. Within the maze there are multiple branching and joining paths which all lead to the health pack. At each fork, the agent randomly chooses a path to follow. The sections between each branch and join are labeled with a different sub-task, for a total of 6 sub-tasks.

**Methodology**
The policy networks follow the same architecture in both domains. The input is a single 60 x 80 x 3 image given by the game pixels from VizDoom, followed by 3 convolutional layers, an LSTM layer, and a fully connected layer. We set the number of predicted clusters (for CSD and K-means) to 8 in both domains. Similarly, the autoencoder has 3 convolutional/deconvolutional layers.

We sample trajectories from expert policies where we have ground truth labels for the different sub-tasks. To train the LSTM architecture, we construct a minibatch by randomly selecting 8 trajectories within our dataset and take subsequences of length 32 from each trajectory. We sequentially continue to feed in the next subsequences of the corresponding trajectories, and randomly select a new trajectory when one trajectory has been fully consumed. Note that there may be an imbalance in number among the different type of constraints. In all of our experiments we set the weights of all constraints to 1 and average over 10 runs. Both domains use 100 train and 25 test trajectories.

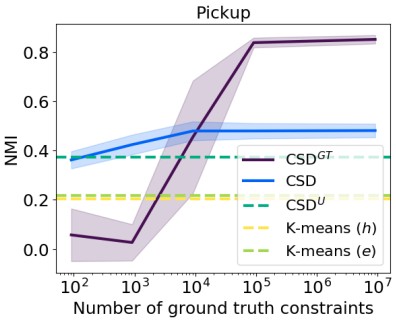 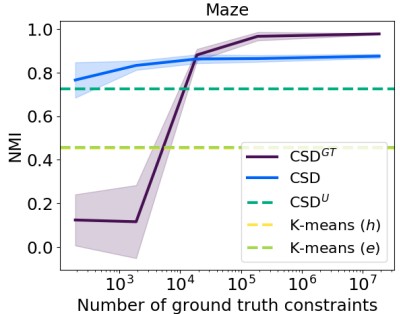

Figure 1: Results for two visual tasks. Performance is measured in normalized mutual information.

| Constraint | Accuracy | | Coverage | |
|---|---|---|---|---|
| | Pickup | Maze | Pickup | Maze |
| $\mathbf{c}^{sch}$ | 66.6% | 72.6% | 100% | 100% |
| $\mathbf{c}^{sce}$ | 69.2% | 75.3% | 100% | 100% |
| $\mathbf{c}^{tc}$ | 72.4% | 95.8% | 10.4% | 11.2% |

Table 1: Statistics over the constraints used by the different approaches.

We compare CSD, which utilizes unsupervised constraints and limited ground truth, to several baselines: 1) clustering with *K-means* using either features $\mathbf{h}$ or $\mathbf{e}$, 2) learning with only the unsupervised constraints ($\text{CSD}^U$), and 3) learning with only limited ground truth constraints ($\text{CSD}^{GT}$).

We measure performance using normalized mutual information (NMI) compared to the ground truth sub-task labels.

**Discussion**
In both domains (Maze and Pickup), shown in Figure 1, our approach with only unsupervised constraints $\text{CSD}^U$ is able to outperform the *K-means* baseline with both features from the policy $\mathbf{h}$ as well as those generated from the autoencoder $\mathbf{e}$. Furthermore, CSD is able to outperform $\text{CSD}^{GT}$ with few ground truth constraints in both domains. This demonstrates the utility of our method for generating weak constraints and combining them via constrained clustering for sub-task discovery. Note that with a sufficient amount of ground truth constraints, $\text{CSD}^{GT}$ outperforms CSD. This may be due to the fact that CSD is combining different types of potentially noisy constraints with a static weight. Dynamically selecting this trade-off is an area for future work.

Next, we examine the quality of the unsupervised constraints by computing the accuracy and coverage (averaged over batches during training), shown in Table 1. Accuracy corresponds to the number of correct constraints out of all constraints generated. Coverage corresponds to the number of generated constraints over all possible constraints. Across the two domains, each of the different types of constraints are better than random, suggesting there is weak information that can be utilized. The quality of the different constraint types varies across domains, suggesting that each type of constraint is capturing a different type of information that CSD can leverage for sub-task discovery.

CONCLUSION AND FUTURE WORK

We have presented Constrained Sub-task Discovery, a semi-supervised strategy for discovering sub-tasks utilizing unlabeled expert trajectories. First, supervised imitation learning is used to approximate the expert policy from trajectories. Using that policy, we extract two types of weak constraints which capture local and global information and are used in constrained clustering. We also learn another set of features through learning an autoencoder directly over the visual states. We demonstrate its improved performance over the baselines in both visual domains. Our method is general in the sense that any approach for feature generation (imitation learning) or constrained clustering can be used. Its key strength is that it does not require any sub-task labels nor the ability to simulate the domain, but can effectively incorporate limited labeled information.

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
