# OpenReview forum: "Sub-Task Discovery with Limited Supervision: A Constrained Clustering Approach"
_ICLR.cc/2019/Workshop/LLD — LLD 2019_

### Official Review · AnonReviewer2 · 2019-04-07
**motivation and methodology could use more substantiation**

**Rating:** 2
**Confidence:** 2

**Review:**

The authors propose a semi-supervised clustering approach to discover subtasks for sequential decision-making. The authors formulate local temporal constraints using entropy with a sliding window over a policy. Global, state-based constraints are determined using expert policy features from imitation learning and a visual autoencoder representation. The authors validate their approach on two VizDoom tasks, Pickup and Maze, treating specific actions as different sub-tasks.

+ Shows clear comparison with ablations of different constraints.
- The problem could be better motivated. There seems to be a gap between the initial problem, sequential decision making, and the approach of constrained clustering. It is not obvious what the “labeling and sample generation requirements” are.
- How will sub-tasks be used in the downstream decision-making? Along those lines, is there an empirical way to measure the downstream efficacy of this approach?
- The formulation of the problem was hard to follow. How can the constraints be used to make decisions? Can you clarify the difference between ground truth labels of sub-tasks and different ground truth constraints?
- “... learns from limited (or no) labeled data by generating weak constraints that do not require sub-task labels or access to a simulator for evaluation.” Is there a trade-off study for none to some limited labels?

While the paper’s motivation appears suggests a no/limited-label approach, it is not evident how the clustering approach is effective for the downstream task.

---

### Official Review · AnonReviewer1 · 2019-04-08
**Well-motivated, well-written paper**

**Rating:** 4
**Confidence:** 1

**Review:**

The authors propose a semi-supervised strategy for sequential decision-making. They initially use supervised imitation learning to approximate the expert policy; then they extract features through weak constraints (temporal and global) from this policy. In addition, they use an autoencoder to complement the aforementioned features.

+ well-written paper
+ well-motivated with clear ideas
- could elaborate more on the differences with the literature
- stronger ablation study about each constraint's contribution should be performed


Some additional question that were not clear while studying the paper:
1) Why do the authors use 8 clusters? What's the intuition behind this number?
2) What are the exact training/implementation details? For instance, how many episodes did their method require?
3) The authors mention that there might be an imbalance among the different types of constraints. Could they elaborate on that?
4) Why are the data with high uncertainty identified as useful sub-goals?

---

### Decision · Program_Chairs · 2019-04-16
**Acceptance Decision**

Accept